# Comparative analysis of temperature effect on bandgap characteristics in 1D phononic crystals: Periodic versus quasiperiodic structures

Farhad Javanpour Heravi[1], Hussein A. Elsayed[1]*, Ali Hajjiah[2], May Bin-Jumah[3], Haifa A. Alqhtani[3], Mostafa R. Abukhadra[4]*, Emad Solouma[5], Suryakanta Nayak[6], Wail Al Zoubi[7], Ahmed Mehaney[1]

1 Photonic and Phononic Crystals Lab., Physics Department, Faculty of Science, Beni-Suef University, Beni-Suef, Egypt, 2 Department of Electrical Engineering, College of Engineering and Petroleum, Kuwait University, Kuwait City, Kuwait, 3 Department of Biology, College of Science, Princess Nourah bint Abdulrahman University, Riyadh, Saudi Arabia, 4 Geosciences Department, College of Science, United Arab Emirates University, Al Ain, United Arab Emirates, 5 Department of Mathematics and Statistics, College of Science, Imam Mohammad Ibn Saud Islamic University (IMSIU), Riyadh, Saudi Arabia, 6 Department of Computer Science and Engineering, Biju Patnaik University of Technology, Odisha, India, 7 Materials Electrochemistry Laboratory, School of Materials Science and Engineering, Yeungnam University, Gyeongsan, Republic of Korea

* drhussien85sc@gmail.com (HAE); m.abdelwahab@uaeu.ac.ae (MRA)

## Abstract

Phononic crystal-based sensors have emerged as highly promising platforms for precise temperature monitoring due to their ability to manipulate acoustic wave propagation through engineered bandgaps. In this work, a 1D phononic crystal composed of alternating layers of tungsten and polycrystalline silicon is systematically investigated in both periodic and quasiperiodic configurations. The study aims to comparatively evaluate periodic and quasiperiodic architectures- including Fibonacci, Thue-Morse, double-periodic, and Cantor sequences, to identify an optimal structural arrangement that maximizes bandgap width and enhances sensing performance. The simulation upshots revealed that the Fibonacci quasiperiodic configuration exhibits the widest Phononic band gap, reaching $18 \times 10^6$ Hz at an operating temperature of 373 K. Meanwhile, the sensor performance is assessed in terms of temperature sensitivity, where the periodic structure demonstrates a stable and linear response over the investigated temperature range, with a maximum sensitivity of 62.5 Hz/K at 373 K. To evaluate practical feasibility, fabrication tolerances are incorporated by considering up to 5% deviations and material property disorders. Additionally, Monte Carlo simulations are employed to analyze the robustness of the transmission spectrum under such uncertainties. In this regard, the investigated results highlight the trade-off between enhanced bandgap characteristics in quasiperiodic structures and the superior stability of periodic configurations, providing valuable insights for the design of high-performance phononic crystal sensors.

**Data availability statement:** All relevant data are within the manuscript and its Supporting information files.

**Funding:** The authors acknowledge Princess Nourah bint Abdulrahman University Researchers Supporting Project number (PNURSP2026R737), Princess Nourah bint Abdulrahman University, Riyadh, Saudi Arabia. The funders had no role in study design, data collection and analysis, decision to publish, or preparation of the manuscript.

**Competing interests:** The authors have declared that no competing interests exist.

# 1. Introduction

The advent of phononic crystal (PnC) has created new opportunities for manipulating the propagation of acoustic waves, similar to the established domains of photonic and electronic band gap materials [1]. PnCs are engineered periodic structures composed of stacked unit cells, each featuring two or more materials in one, two or three dimensions that exhibit a significant contrast in their elastic and acoustic characteristics [2,3]. These structures hold a great potential due to their distinctive properties, particularly the formation of phononic band gap (PnBG), that prevent the transmission of acoustic waves at designated frequencies [2–6]. Notably, the emergence of such stop bands is essentially related to the significant contrast in the elastic and acoustic properties of these designed structures. Therefore, the ability of these designed structures to effectively manage and manipulate acoustic waves renders them highly valuable in various physical, biomedical, and chemical fields. In the context of sensors and detections, PnCs have been utilized as fluidic sensors [7–9], the detection of hazardous metallic oxides in water [10,11] and the monitoring of chemical liquid concentrations [12,13]. Over the last decade, the researchers have shown a keen interest towards the employment of PnCs in many electronic and engineering applications such as, acoustic diodes [14,15], high-resolution acoustic imaging devices [16], and acoustic mirrors [17]. Interestingly, the mainstay of these sensors is essentially based on the tunable characteristics of the PnBG and the resonant modes that could appear inside it as well [18]. In this regard, the introduction of a defect layer within an otherwise perfect PnC configuration induces localized resonant modes inside the PnBG, which are highly sensitive to some external perturbations and therefore could be promising for sensing applications. However, the deviations from strict periodicity in layered PnC structure can similarly modify the dispersion characteristics, resulting in the appearance of resonance-like states within the bandgap even in the absence of an explicit defect layer. In this context, quasiperiodic PnC designs provide a fundamental framework for modelling structural irregularities, as they inherently introduce deterministic aperiodicity that enables bandgap engineering and localized mode formation.

Therefore, quasi-periodic PnCs have attracted significant attention due to their unique wave manipulation capabilities arising from aperiodic order and long-range correlations. Unlike conventional periodic PnCs, quasi-periodic PnCs exhibit richer spectral characteristics, including multiple and isotropic bandgaps, enhanced wave localization, and fractal-like dispersion properties [19,20]. These features enable superior control over acoustic and elastic wave propagation. Existing studies have explored Fibonacci, Thue-Morse, and Penrose-based structures for applications such as vibration isolation, acoustic filtering, and waveguiding [21]. Additionally, the quasiperiodic arrangements can generate more intricate and adjustable PnBGs compared to the periodic structures [19,20,22–24]. Despite the apparent irregularity of quasiperiodic structures, which might lead one to assume their performance is minimal and impractical, these structures actually conform to specific regularities and exhibit satisfactory performance as sensors [25,26]. It's worth noting that the quasiperiodic PnC designs introduce multiple interference effects due to their deterministic

aperiodicity, leading to modifying the acoustic wave propagation, the formation of multiple PnBGs, and the emergence of localized resonant modes. These features make quasiperiodic structures particularly attractive for phononic sensor design. Therefore, such architectures have been widely explored in chemical, thermal, physical, and biomedical sensing applications [27–32]. Since acoustic wave velocity is strongly temperature-dependent and directly governs transmission characteristics, temperature effects play a critical role in phononic-based sensing. Consequently, PnCs are especially suited for thermal sensing applications, offering an enhanced sensitivity to temperature-induced variations in material and wave propagation properties similar to what have been achieved for different sensing applications [33–49].

Meanwhile, A. Almawgani et al, reported a Fibonacci-based quasi-periodic PnC sensor, which achieves a superior bio-liquid sensing performance. The authors analyzed the transmission spectrum employing the transfer matrix method and accomplished a maximum sensitivity of 959 MHz besides high quality factor of 6947.059 and figure of merit of 323.5 for detecting different concentrations of NaI solutions [50]. A G Sayed et al. presented a theoretical design of a highly sensitive 1D quasi-periodic PnC fluidic sensor, where a defect-layer-induced resonant mode enables the detection of $Pb(NO_3)_2$ concentrations, with the double-period sequence achieving maximum sensitivity of 502.6 Hz/ppm [25]. A. Almawgani and his team demonstrated a robust quasiperiodic topological PnC sensor incorporating Fano resonance to enhance the stability against the fabrication imperfections and enable high-performance temperature sensing of propanol. Additionally, the Fibonacci-based design demonstrates superior performance, achieving high sensitivity, quality factor, and reliability over a wide temperature range (160–240 °C) [51]. S E Zaki et al investigated an ultra-sensitive greenhouse gas sensors based on periodic and quasi-periodic PnCs using $Pt/PtS_2$ composites, where sharp Fano resonance modes significantly enhance the sensing performance. The optimized Fibonacci structure achieved the best results, delivering high sensitivity, quality factor, and figure of merit towards the detection of $CH_4$ along with strong temperature-dependent performance [52].

This work explicitly explores the impact of temperature on the performance of PnC structure, focusing on the effectiveness of both periodic and quasiperiodic type PnCs. The novelty of this work lies in the comprehensive comparison of periodic and multiple quasiperiodic one-dimensional PnC configurations for sensing applications using an identical tungsten–polycrystalline silicon platform. Unlike prior studies focusing on single architectures, this work systematically evaluates Fibonacci, Thue-Morse, double-periodic, and Cantor sequences to elucidate their bandgap and sensing characteristics. The identification of the Fibonacci structure as providing maximum bandgap enhancement, alongside the demonstration of superior thermal stability and linear sensitivity in periodic designs, is a key contribution. Specifically, the Fibonacci quasiperiodic structure achieves a maximum bandgap width of $18.094 \times 10^6$ Hz, which represents a $\sim 52\%$ increase compared to the periodic structure ($11.889 \times 10^6$ Hz). In terms of sensing performance, a peak sensitivity of $-1175$ Hz/K is obtained for the Fibonacci configuration, which is significantly higher than the periodic case ($-584$ Hz/K) and exceeds several previously reported values for temperature-based PnC sensors. Furthermore, the inclusion of fabrication tolerances and Monte Carlo-based robustness analysis offers practical insight into real-world sensor reliability.

## 2. Structure analysis and mathematical modelling

### 2.1. Proposed structure

The proposed quasi-periodic structure is realized with three-unit cells, each incorporating tungsten and polycrystalline-silicon, as shown in Fig 1. These unit cells are configured as, (*Tungsten/Polycrystalline Silicon*)$^3$. The thicknesses of the layers are referred to as $a_1$ and $a_2$ respectively, with a lattice constant, $a = a_1 + a_2$. Here, the alternating tungsten and polycrystalline silicon layers in the structure give rise to the emergence of acoustic bandgaps owing to the strong contrast in their elastic and mass density properties. Tungsten possesses high density and elastic modulus, while polycrystalline silicon exhibits comparatively lower density and stiffness. This periodic modulation in acoustic impedance causes multiple reflections and Bragg scattering of incident acoustic waves at the layer interfaces. Moreover, both these materials exhibit well-characterized and stable temperature-dependent properties, making them appropriate for temperature sensing

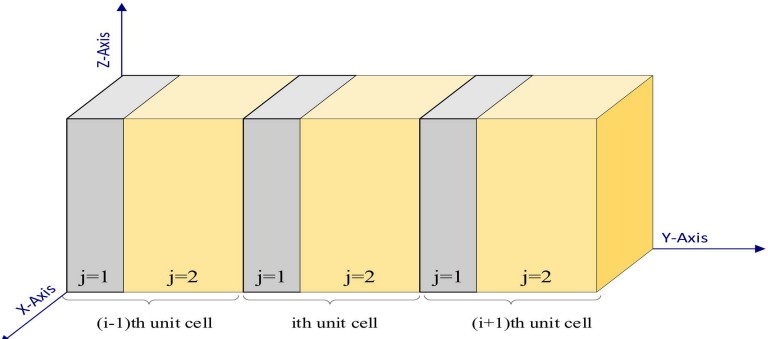

**Fig 1. Schematic representation of the ideal structure consisting of three-unit cells at a temperature of 293 K.**

applications. When the lattice periodicity is comparable to the acoustic wavelength, destructive interference suppresses wave propagation over specific frequency ranges, leading to the formation of PnBG, that prevent acoustic energy transmission. In the analysis of the transmission spectrum, Young's modulus and density are critical parameters, as illustrated in Fig 2. At 293K, Young's modulus for tungsten and polycrystalline silicon are considered as 410 GPa and 163.8 GPa respectively, whereas the density of these materials is taken as 19260 kg/m³ and 2324 kg/m³, respectively. Then, Poisson's ratios for tungsten and polycrystalline silicon are specified as 0.26 and 0.22, respectively. Interestingly, the numerical data of tungsten and polycrystalline silicon in Fig 2 we fitted based on the experimental data found in references [53–55].

## 2.2. Mathematical formulations

The periodic variation in acoustic impedance across the layered medium induces successive partial reflections at each interface, giving rise to coherent multiple scattering and Bragg interference of the incident acoustic waves [34,35]. When the structural periodicity satisfies the Bragg condition, i.e., the wavelength becomes comparable to the double of the lattice constant, thus the constructive interference of reflected waves and destructive interference of transmitted waves occur. This response through the attenuation of propagating modes within specific frequency intervals, thereby could be

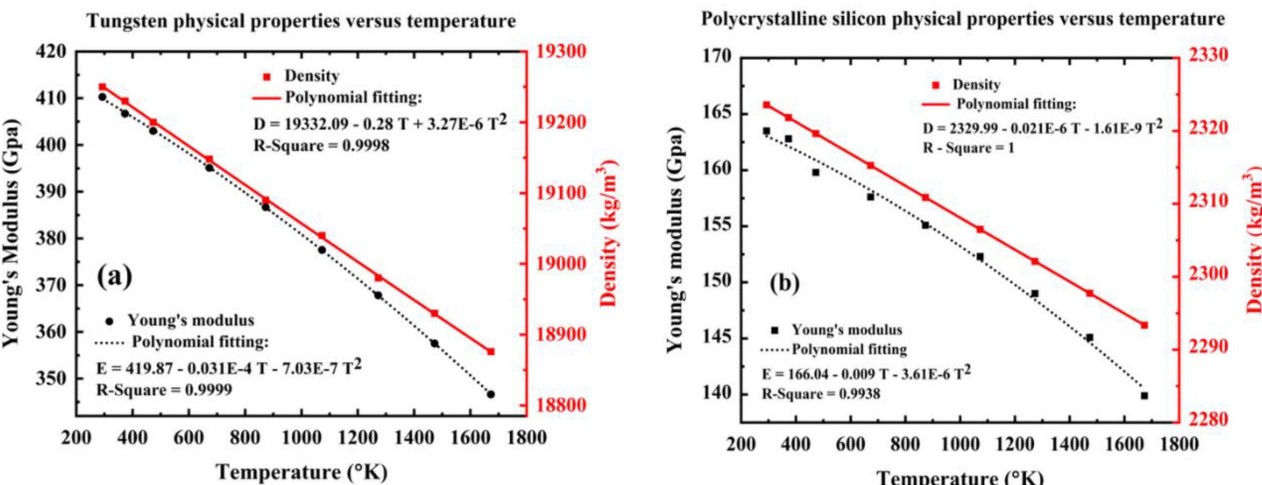

**Fig 2. Variation of the intrinsic properties of materials with respect to temperature (a) tungsten material (b) polycrystalline silicon material.**

effective in the emergence of the PnBGs [36,37]. Within these forbidden bands, the propagation constant becomes complex, leading to evanescent decay of acoustic waves and effectively inhibiting energy transmission through the structure [38]. The characteristics of the PnBGs are analyzed in the transmission spectrum of the structure. In the simulation, the boundary conditions are defined by enforcing continuity of both displacement and normal stress at each material interface, by maintain consistency with the linear electrodynamic theory. The external boundaries are modelled as semi-infinite media, thereby eliminating artificial reflections, and ensuring physically realistic wave propagation.

To investigate the transmission spectrum, it is essential to understand the characteristics of acoustic waves propagating through the layered media. Consequently, the initial step involves solving the acoustic wave equation and defining the state vectors on both the left and right sides of each unit cell. Acoustic waves undergo scattering at each interface between distinct layers. By considering the continuity of the waveforms on either side of each interface and the effects of multiple scattering, the superposition principle manifests between the incident and reflected components of the acoustic wave. Meanwhile, the proposed wave equation that encompasses these effects can be expressed in the following form,

$$\rho \ddot{\varphi} = \frac{\partial}{\partial x}(\sigma) + f$$

(1)

Where, $\rho = \rho(T)$ refers to the materials density, which is function as a temperature, $\varphi = \varphi(x, t)$ displacement, $\sigma = \sigma(x, t)$, and $f(x, t)$ stand for stress and external force respectively. Here, Equation (1) shows the scalar electrodynamic wave equation formulated under the assumption of longitudinal wave propagation normal to the layered interfaces. In this study, the transverse modes are neglected due to the dominance of longitudinal waves in layered media. Further, the stress-displacement relations are simplified by using the isotropic linear elasticity. The boundary conditions assume continuity of both displacement and stress at each interface, while semi-infinite, non-reflecting media are considered at the outer boundaries. These assumptions are consistent with the transfer matrix framework adopted in this research. Therefore, the proposed solution for the considered equation can be written as follows:

$$\varphi(x.t) = (Ae^{ikx} + Be^{-ikx})e^{i\omega t}$$

(2)

Where, $A$ and $B$ denote the amplitudes, $k = \frac{\omega}{C_j}$ is the wave vector, $\omega$ is the angular frequency, and $C_j$ is the longitudinal wave velocity in each layer. For the longitudinal acoustic wave, which is incident normally to the structure, Equation (1) can be rewritten as the following:

$$\frac{\partial^2 \varphi}{\partial x^2} = \frac{1}{C_L^2} \frac{\partial^2 \varphi}{\partial t^2}$$

(3)

$$C_L = \sqrt{\frac{(\lambda + 2\mu)}{\rho}}$$

(4)

Where, $\lambda$ and $\mu$ stand for Lame parameters, where $\mu$ specifically represents the shear modulus. As previously mentioned, the presence of multiple scatters and the continuity of the waveform at each interface leads to the occurrence of superposition between the incident and reflected waves. Therefore, the continuity condition can be expressed in the following manner:

$$v_{jR}^{(i)} = T_i v_{jL}^{(i)}$$

(5)

$$v_{1L}^{(i)} = v_{2R}^{(i-1)}$$

(6)

In this context, (i) and (i-1) refer to the $i^{th}$ and $(i-1)^{th}$ unit cells, respectively. Then, we have introduced the term $T_i$ to identify the transfer matrix that connects the left and right sides of each unit cell, and it can be defined as follows:

$$T_i = T_2 T_1 \tag{7}$$

Here, $v_{jR}^{(i)}$ and $v_{jL}^{(i)}$ serve as a state vector to describe the properties of the interacting waves at the entrance of crystals and after traveling throughout the crystal. Based on state vectors, the continuity of wave at every interface of PnC design can be simulate by implementing transfer matrix method (TMM). Therefore, the state vector can be written as:

$$v_{jR}^{(i)} = \left\{ \varphi_{jR}^{(i)}, \overline{a_1} \tau_{jR}^{(i)} \right\}^T, \quad v_{jL}^{(i)} = \left\{ \varphi_{jL}^{(i)}, \overline{a_1} \tau_{jL}^{(i)} \right\}^T \tag{8}$$

$$T = T_n T_{n-1} \ldots T_i \ldots T_1 \quad (i = 1, 2, 3, \ldots, n) \tag{9}$$

Where, $\tau_{jR}$ and $\tau_{jL}$ are the dimensionless components of stress at the left and right side of $i^{th}$ unit cell. In a binary crystal, each unit cell is composed of two materials therefore $j = 1, 2$. By implementing state vectors in each interface and combining Equation (2) with Equation (8), we can provide a simple matrix form. For the right side of the interface, we have:

$$\varphi_{jR}\left(x_{i+1}\right) = \left(A_j \exp\left(-ikx_{i+1}\right) + B_j \exp\left(ikx_{i+1}\right)\right) \tag{10}$$

$$\tau_{xzjR}\left(x_{i+1}\right) = \mu_j^{(i)} \frac{\partial \varphi_{jR}^{(i)}}{\partial x_{i+1}} = \mu_j \left(-iA_j k \exp\left(-ikx_{i+1}\right) + iB_j k \exp\left(ikx_{i+1}\right)\right) \tag{11}$$

In contrast, for the left side of the interface, we have:

$$\varphi_{jL}\left(x_i\right) = \left(A_j \exp\left(-ikx_i\right) + B_j \exp\left(ikx_i\right)\right) \tag{12}$$

$$\tau_{xzjL}\left(x_i\right) = \mu_j^{(i)} \frac{\partial \varphi_{jL}^{(i)}}{\partial x_j} = \mu_j \left(-iA_j k \exp\left(-ikx_i\right) + iB_j k \exp\left(ikx_i\right)\right) \tag{13}$$

Then, Equations (10), (11), (12), and (13) for the right and left sides of the interface can be expressed together in the following matrix formulism [39]:

$$\begin{bmatrix} \varphi_{jL}(x_i) \\ \tau_{jL}(x_i) \end{bmatrix} = \begin{bmatrix} \exp(-ikx_i) & \exp(ikx_i) \\ -i\mu_j k\exp(-ikx_i) & i\mu_j k\exp(ikx_i) \end{bmatrix} \begin{bmatrix} A_j \\ B_j \end{bmatrix} \tag{14}$$

$$\begin{bmatrix} \varphi_{jR}(x_{i+1}) \\ \tau_{jR}(x_{i+1}) \end{bmatrix} = \begin{bmatrix} \exp(-ikx_{i+1}) & \exp(ikx_{i+1}) \\ -i\mu_j k\exp(-ikx_{i+1}) & i\mu_j k\exp(ikx_{i+1}) \end{bmatrix} \begin{bmatrix} A_j \\ B_j \end{bmatrix} \tag{15}$$

Then, by investigating the values of $A_j$ and $B_j$, the transfer matrix for each sublayer can be define as the follow [34,35],

$$T_j = \begin{bmatrix} \cos\left(k_j x_j\right) & \frac{-i\sin(k_j)}{\alpha_j} \\ -i\alpha_j \sin\left(k_j x_j\right) & \cos\left(k_j x_j\right) \end{bmatrix} \tag{16}$$

The terms $x_{i+1}$ and $x_i$ refer to the displacement between adjacent unit cells, $k_j = \sqrt{(\frac{\omega}{c_{Lj}})^2 - (kx)^2}$, $kx = \frac{\omega}{c_{Lj}} \sin \theta$, and $\alpha_j = ik_j\mu_j$ such that $\mu_j$ alludes to the shear modulus component. Then, the transmission coefficient can be found by using the elements of the transfer matrix $T_j$, which is stated as [40],

$$t = \frac{2\gamma_O(T_{11}T_{22} - T_{12}T_{21})}{\gamma_O(T_{11} - \gamma_e T_{21}) - (T_{12} - \gamma_e T_{22})} \tag{17}$$

Where, $T_{11}$, $T_{12}$, $T_{21}$ and $T_{22}$ denote the elements of the total transfer matrix. Lastly, transmittance (T) of the proposed structure can be numerically expressed as,

$$T = \frac{\gamma_e}{\gamma_0}\left|t^2\right| \tag{18}$$

## 3. Results and discussions

The periodic (perfect) structure is adopted as a reference baseline for assessing the performance of quasiperiodic configurations. Accordingly, the structure is simulated with respect to some key parameters like layer thickness and angle of wave incidence. In this research, the angle of incidence refers to the angle between an externally excited plane acoustic wave and the normal to the layered interfaces. The structure is modelled as a stratified multilayer medium, where oblique incidence modifies the normal component of the wavevector, thereby influencing the Bragg's condition and bandgap formation. So, the incidence angle plays a critical role in governing the bandgap characteristics of the PnC structures. Fig 2a and b illustrate the temperature-dependent variations of Young's modulus and density for tungsten and polycrystalline silicon, respectively, which are the constituent materials of the proposed PnC unit cell. For tungsten, both Young's modulus and density exhibit a monotonic decrease with increasing temperature, reflecting enhanced lattice vibrations and thermal expansion effects. The polynomial fitting accurately captures this trend, thereby achieving a high coefficient of determination ($R^2 \approx 0.999$). Similarly, polycrystalline silicon shows a pronounced decrement in the elastic stiffness and mass density with respect to increase in the temperature. The slightly lower $R^2$ values still confirm a good agreement between the fitted curves and data. These temperature-dependent material properties directly influence both acoustic wave velocity and impedance contrast within the PnC structure, thereby modulating the position and width of PnBG. Consequently, incorporating such realistic thermal dispersion is essential for accurate prediction of PnC performance under elevated temperature operating conditions.

### 3.1. Perfect 1D Periodic PnC

Next, we analyze a perfect 1D PnC periodic configuration consisting of (Tungsten/Polycrystalline Silicon)³ with primary thicknesses of $a_1 = 1mm$ and $a_2 = 2mm$, respectively, to determine the optimal incidence angle associated with the broader bandgaps, as illustrated in Fig 3a–d. At normal incidence ($\theta = 0°$, Fig 3a, the structure exhibits well-defined PnBGs characterized by near-zero transmission regions, arising from strong Bragg scattering due to the periodic acoustic impedance mismatch between tungsten and polycrystalline silicon layers. As the incident angle increases to 45° (Fig 3b) and 60° (Fig 3c), noticeable modifications in the spectral response are observed, including the shift and expansion of stopbands. This behavior is attributed to the angular dependence of the effective wavevector component normal to the layers, which alters the Bragg condition and enhances mode coupling within the periodic stack. At a highly oblique incidence of 85° (Fig 3d), the bandgaps become significantly wider and more pronounced, indicating stronger suppression of acoustic wave propagation. From Fig 3, it is concluded that an increase in the incident angle correlates with a wider bandgap. In the context of PnC, the optimal incident angle that yields satisfactory results is $\theta = 78°$. As the angle nears grazing incidence ($\theta \rightarrow 90°$), wave propagation experiences significant attenuation, heightened sensitivity to surface irregularities, and complications related to total internal reflection. Furthermore, from a theoretical standpoint, this angular range adequately

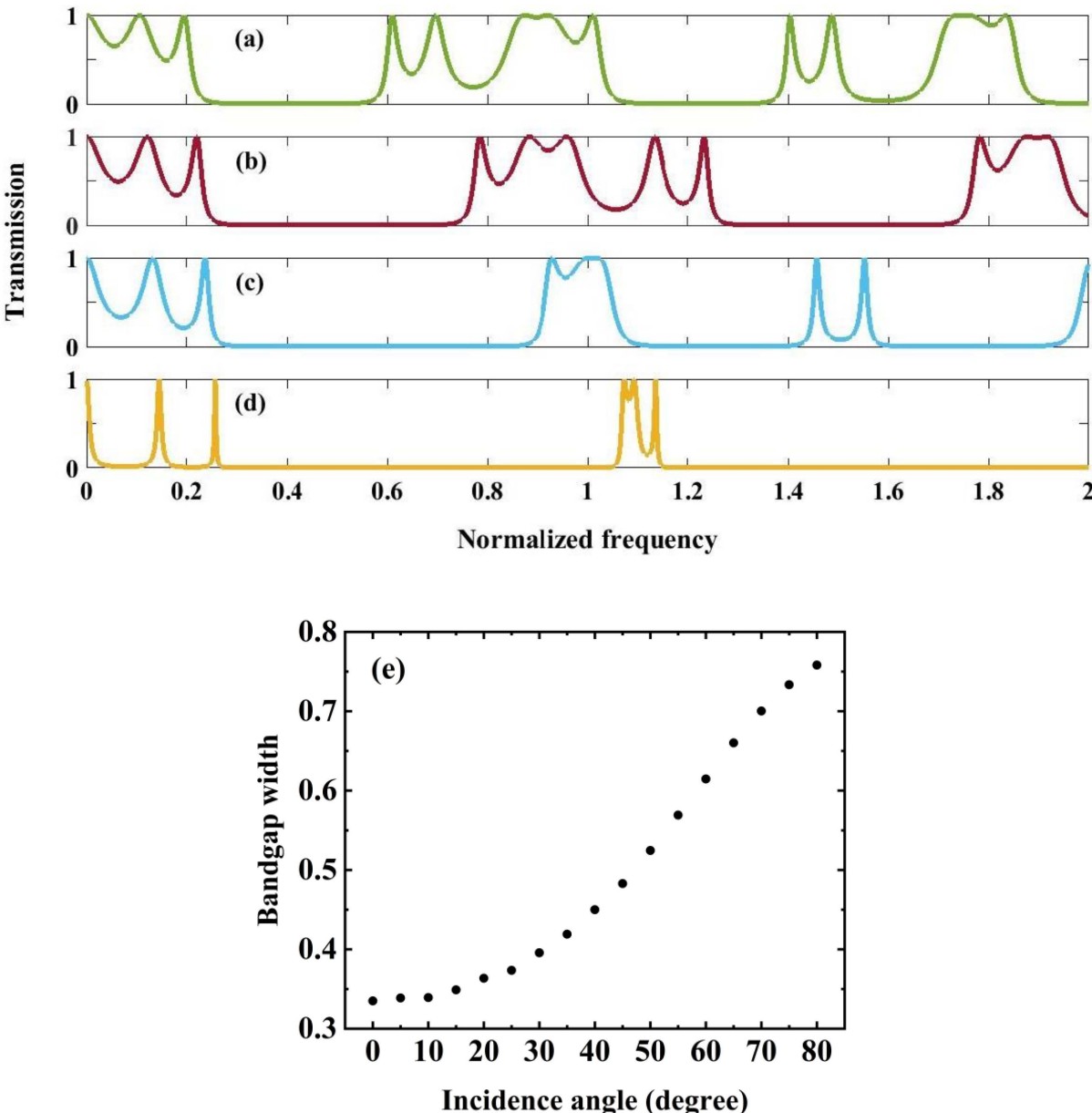

**Fig 3. Demonstrate the transmission spectrum of perfect structure in a (Tungsten/Polycrystalline Silicon)³ configuration under the circumstance of various incident wave angles. (a)** $\theta = 0°$ **(b)** $\theta = 45°$ **(c)** $\theta = 60°$ **(d)** $\theta = 85°$ **(e)** The variation of band gap width versus incident angles from $\theta = 0°$ to $\theta = 80°$.

encompasses the fundamental effects of Bragg scattering, since larger angles yield negligible additional insights regarding the formation of the bandgap or the characteristics of wave propagation. Experimental constraints, including finite aperture effects, detector limitations, and practical challenges in wave generation, impose additional restrictions on the range of usable angles. The angle $\theta = 78°$ is identified as optimal, as it effectively balances these factors, allowing for comprehensive data collection while ensuring accuracy and mitigating issues related to extreme incidence angles. Fig 3e quantitatively summarizes this trend by illustrating the variation of bandgap width as a function of incidence angle from 0° to 85°.

A monotonic increase in bandgap width is observed, confirming that oblique incidence enhances bandgap formation. This angular tunability highlights the critical role of incidence angle in tailoring the filtering characteristics and wave control capabilities of the proposed PnC structure.

Afterwards, we move on to optimize the thickness of the different layers in the (Tungsten/Polycrystalline silicon)[3] configuration. In the entire research, we have maintained the thickness of tungsten layer at a minimum thickness of 1 mm, which is intended to streamline the crystal fabrication process. Subsequently, this layer serves as a foundation for optimizing the thickness of the polycrystalline silicon layer. Fig 4 depicts the variation of the width of the PnBG as a function of polycrystalline silicon layer thickness in the structure. It is observed that an initial increase in thickness leads to a rapid expansion of the bandgap due to enhanced Bragg scattering and improved the acoustic impedance contrast between adjacent layers. Beyond an optimal thickness, the bandgap width saturates, indicating that further geometric scaling provides diminishing returns. This saturation is governed by intrinsic material properties and wave coupling limitations. It is revealed that indicates that the maximum bandgap is associated with a thickness of 9 mm, which is considered as an optimized thickness in this research. The results emphasize the importance of thickness optimization to maximize bandgap width and achieve efficient acoustic wave confinement in the designed PnC structure.

Fig 5 illustrates the normalized transmission spectrum of the perfect (Tungsten/Polycrystalline Silicon)[3] PnC under optimized layer thickness conditions. Numerous distinct passbands with high transmission are observed at low normalized frequencies, followed by a wide frequency region exhibiting near-zero transmission, corresponding to a pronounced PnBG. This broad stopband arises from the strong Bragg scattering induced by the periodic modulation of acoustic impedance between tungsten and polycrystalline silicon layers. At higher normalized frequencies, narrow transmission peaks reappear, indicating the onset of higher-order propagating modes. The well-defined bandgap confirms the effective acoustic wave suppression and demonstrates that the optimized structural parameters enable robust bandgap formation, essential for high-performance sensing applications.

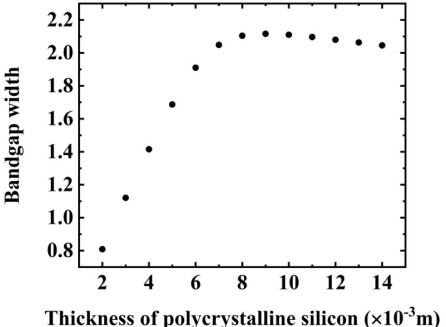

**Fig 4. The correlation between the bandgap width of the (Tungsten/Polycrystalline silicon)³ configuration and the thickness of the polycrystalline silicon layer.**

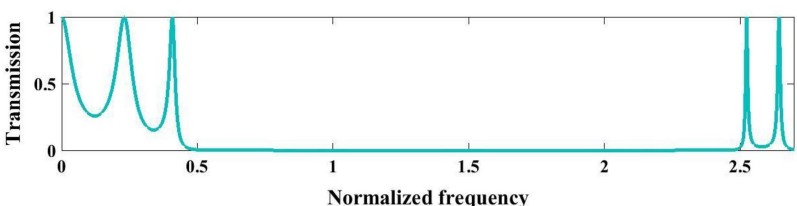

**Fig 5. Transmission spectrum of the perfect structure, with (Tungsten/Polycrystalline silicon)³ configuration under optimal condition.**

## 4. Quasiperiodic structures

Subsequently, a discussion on quasiperiodic structures is provided, comparing them with the periodic structure. It is important to note that the periodic optimal structure serves as a reference point for evaluating the performance of both periodic and quasiperiodic structures under varying temperature conditions. In this work, we investigate various quasiperiodic sequences like Fibonacci, Thue-Morse, double periodic, and Cantor configurations for designing binary PnC structures.

### 4.1. Fibonacci sequence

The Fibonacci sequence, introduced by Leonardo Fibonacci in the 13th century, is defined by a recursive series where each term equals the sum of the preceding two [41]. In quasiperiodic structures, the Fibonacci sequence is implemented using two building blocks, A and B, arranged in such a way that the $n^{th}$ stage of the process $S_n$ is given by the recursive rule for $n \geq 2$, starting with $S_0 = A$ and $S_1 = B$. The Fibonacci sequence generation can be defined as, $S_n = S_{n-1}S_{n-2}$, $S_0 = A$, $S_1 = B$, $S_2 = BA$, $S_3 = BAB$, $S_4 = BABBA$, $S_5 = BABBABAB$ etc. Fig 6 illustrates the normalized transmission spectrum of the Fibonacci $S_3$ quasiperiodic PnC at 293K. From this figure, it is observed that a sharp transmission peak ($T \approx 1$) appears near 0.35, corresponding to a resonant passband. Beyond this, a wide PnBG is observed from 0.5 to 2.4, where transmission remains nearly zero ($T \approx 0$), which confirms a strong Bragg scattering and wave attenuation. Additionally, a highly localized and narrow resonance peak appears at a normalized frequency of 2.5 with transmission approaching unity, indicating a highly localized mode induced by quasi-periodicity. Following this, another stopband persists up to 3.3 with a very low transmission. Finally, the transmission sharply increases again near 3.5, marking the onset of higher-order propagating modes. Overall, the structure demonstrates a broad bandgap with a PnBG width of 1.9, and sharp defect-like resonances, highlighting strong wave localization. These characteristics confirm the ability of Fibonacci quasi-periodicity to tailor bandgap distribution and enhance acoustic wave confinement, demonstrating its potential for advanced sensing applications.

### 4.2. Thue-Morse sequence

The Thue–Morse sequence, introduced by Prouhet in 1851, is an infinite binary sequence generated by recursively appending the Boolean complement of the existing sequence. In layered media design, its recursive definition is essential for accurately constructing quasiperiodic configurations with well-defined structural order and wave propagation characteristics [42,43]. The generation rule of the Thue-morse sequence can be defined as the following,

$$\begin{cases} S_n = S_{n-1}S'_{n-1} \\ S'_n = S'_{n-1}S_{n-1} \end{cases} \quad \text{for } n \geqslant 1$$

If $S_0 = A$ and $S_1 = B$, the sequence can be defined as, $S_0 = A$, $S_1 = B$, $S_2 = ABBA$, $S_3 = ABBABAAB$, $S_4 = ABBABAABBAABABBA$ etc.

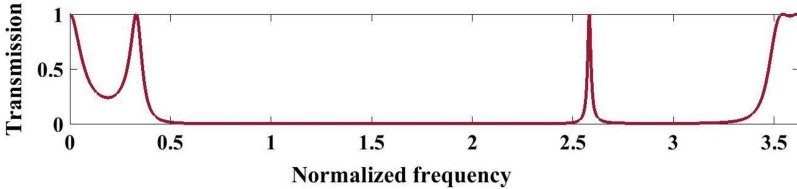

**Fig 6. Analysis of transmission spectrum of the phononic crystal designed according to the three sequence S3 of the Fibonacci series (Tungsten/Polycrystalline silicon/Tungsten) under optimal conditions at a temperature of 293K.**

Fig 7 presents the normalized transmission spectrum of a quasiperiodic PnC designed using the third-order Thue–Morse sequence $S_3$ under the optimized structural parameters at 293K. The spectrum is characterized by extended near-zero transmission regions, indicating the formation of a wide PnBG resulting from the strong multiple scattering and impedance mismatch between the constituent layers. Unlike periodic structures, the Thue–Morse quasi-periodicity introduces a deterministic aperiodic order, leading to the emergence of sharp transmission peaks embedded within the stopband. As perceived from this figure, a wide bandgap extends approximately from 0.4 to 2.5, where transmission is nearly zero (T ≈ 0), indicating strong attenuation. Unlike the Fibonacci case, two sharp localized resonance peaks appear at 2.58 and 2.64 with transmission approaching unity, reflecting coupled localized modes. Beyond this point, the transmission again drops, confirming multiple interference effects and slightly narrower but structured bandgap behavior. The pronounced suppression of transmission over a broad frequency range arises from enhanced wave interference and phase incoherence caused by the non-periodic stacking sequence. Additionally, the appearance of isolated high-transmission peaks at higher normalized frequencies corresponds to higher-order resonances permitted by the quasiperiodic symmetry. Overall, the observed spectral variations highlight the strong influence of Thue–Morse ordering on acoustic wave localization and bandgap engineering in the proposed PnC structure.

## 4.3. Double-periodic sequence

Doubly periodic functions originated from elliptic integrals and were formally introduced by Abel in 1827. Double-periodic sequences exhibit periodicity at two distinct scales and are widely applied in crystallography, signal processing, and wave physics. Their mathematical description enables analysis of multiple bandgaps and enhanced wave control in engineered structures [44–46]. The generation of double-periodic sequences can be defined as,

$$\begin{cases} S_n = S_{n-1}S'_{n-1} \\ S'_n = S_{n-1}S_{n-1} \end{cases} \quad \text{for } n \geqslant 1$$

Considering $S_0 = A$ and $S'_0 = B$ the sequence provides its components as following forms:

$S_0 = A$, $S_1 = AB$, $S_2 = ABAA$, $S_3 = ABAAABAB$, $S_4 = ABAAABABABABAAABAA$, *etc*.

Meanwhile, Fig 8 presents the normalized transmission spectrum of a quasiperiodic PnC designed using the third-order double-periodic sequence under the optimized structural parameters at 293K. A broad near-zero transmission region is observed, indicating the formation of an extended PnBG resulting from the combined effect of two distinct periodic length scales. This dual periodic modulation enhances Bragg scattering and induces complex interference among multiple reflected acoustic waves, leading to strong suppression of wave propagation over a wide frequency range. As depicted in this figure, a wide stopband extends approximately from 0.5 to 2.2 with near-zero transmission, indicating the strong attenuation. A narrow-isolated resonance appears at a normalized frequency of 1.2, suggesting a localized mode within the bandgap. Additional sharp peaks occur around 2.25, 2.58, and 2.7 with high transmission, reflecting multi-scale

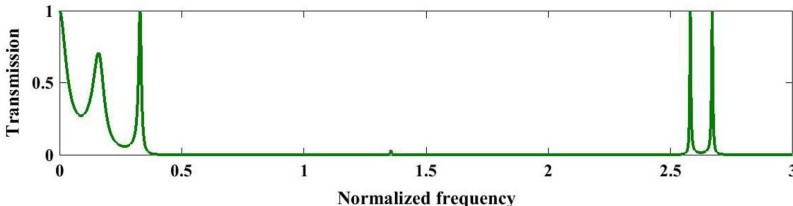

**Fig 7. Illustrates the transmission spectrum of the phononic structure by implementing three sequence S3 of the Thue-Morse series under optimal conditions at a temperature of 293K.**

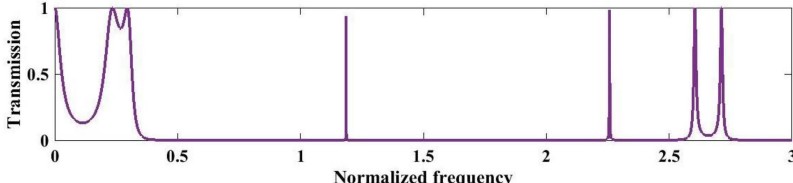

**Fig 8. Analysis of transmission spectrum of the third sequence S3 of the double-periodic series PnC under optimal conditions at a temperature of 293K.**

interference due to the dual periodicity. The appearance of isolated high-transmission spikes at higher normalized frequencies is associated with higher-order resonances permitted by the multi-scale structural symmetry. In contrast to single-periodic structures, the double-periodic arrangement introduces additional sharp transmission peaks within the stopband, which correspond to localized resonant modes arising from constructive interference between the superimposed periodicities. These spectral variations demonstrate that double-periodic quasi-periodicity provides enhanced flexibility in tailoring bandgap distribution and resonance characteristics, making it highly suitable for advanced acoustic filtering, waveguiding, and sensing applications.

## 4.4. Cantor sequence

The Cantor set, introduced by Smith and formalized by Cantor, is a deterministic fractal constructed by iteratively removing the middle third of a line segment. The triadic Cantor sequence exemplifies an uncountable set with zero Lebesgue measure and is fundamental to fractal geometry and wave-structured systems [47,48]. In this work, we focus on the outbound Cantor sequence, where the $n^{th}$ stage is defined by the expression [49,56]:

$$S_n = S_{n-1}B_nS_{n-1}$$

$B_n$ is defined based on its thickness as follows,

$$d_{B_n} = 3^{n-1}d_{B_1}$$

If $B_1 = B$ the sequence generation can be defined as, $S_0 = A$, $S_1 = ABA$, $S_2 = ABABBBABA$,
$S_3 = ABABBBABABBBBBBBBBABABBBBABA$, $S_4 = S_3 3^3 BS_3$ *etc.*

Fig 9 illustrates the normalized transmission spectrum of the quasi-periodic PnC designed using the third-order Cantor sequence under the optimized conditions at 293 K. The spectrum exhibits multiple fragmented stopbands with intermittent sharp transmission peaks, reflecting the fractal and self-similar nature of the Cantor geometry. These spectral variations

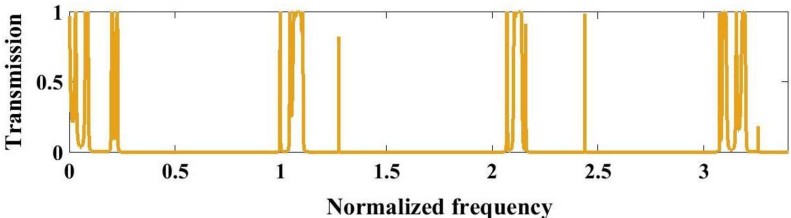

**Fig 9. Analysis of transmission spectrum of the three sequence S3 of the Cantor set under optimal conditions at a temperature of 293K.**

arise from hierarchical Bragg scattering at different length scales introduced by the iterative removal process. A broad suppressed region extends roughly from 0.25 to 1.0 with near-zero transmission. Beyond this, several isolated resonances appear at normalized frequency from 1.0–1.3 and 2.0–2.3. In the higher frequency range (~3.0–3.3), dense clusters of sharp transmission peaks are observed. These multiple narrow passbands and stopbands confirm strong fractal-induced localization and multi-scale bandgap formation across the spectrum. Such complex bandgap characteristics demonstrate the ability of Cantor-based quasiperiodic structures to realize multi-band filtering and enhanced control over acoustic wave propagation.

## 5. Temperature effect

The temperature plays a crucial role in sensing performance of PnC structure. As, mentioned in Fig 2, for both materials polycrystalline silicon and tungsten, young's modulus and density decrease by the enhancement of temperature. Therefore, the necessity of color map to clarify the variation of band gap versus temperature is perceived. Fig 10 illustrates the temperature-dependent transmission characteristics of the perfect 1D PnC with (Tungsten/Polycrystalline Silicon)³ configuration. In Fig 10a, the 3D transmission map depicts the combined influence of normalized frequency and temperature on acoustic wave propagation. As the temperature increases, a gradual shift and modulation of the transmission features are observed, which can be attributed to the temperature-induced variation of elastic constants and mass density of the constituent materials. These changes alter both acoustic impedance contrast and effective wave velocity, thereby influencing the bandgap formation. Fig 10b presents a 2D colormap plot highlighting the evolution of the PnBG with respect to temperature and normalized frequency. Broad regions of near-zero transmission persist over a wide temperature range, indicating robust bandgap stability against thermal fluctuations. The slight frequency drift of the band edges with temperature reflects the thermal softening of materials, which modifies the Bragg scattering condition. Fig 10c shows the transmission spectrum as a function of normalized frequency at a representative temperature, clearly identifying passbands and well-defined stopbands. The pronounced suppression of transmission within the bandgap confirms strong Bragg reflection arising from periodic impedance mismatch.

Fig 11 demonstrates the temperature-dependent transmission characteristics of a Fibonacci quasiperiodic PnC with a three-sequence configuration (Tungsten/Polycrystalline Silicon/Tungsten). In Fig 11a, a 3D view of the transmission map is presented, where multiple pronounced stopbands and passbands are perceived, whose spectral positions vary with both normalized frequency and temperature. The quasiperiodic Fibonacci ordering facilitates an enhanced scattering and wave interference, leading to wider and more complex bandgap structures compared to periodic counterparts. Fig 11b presents a contour representation of transmission, where a formation of bandgap width with varying temperature can be clearly observed. The persistence of near-zero transmission regions over a wide temperature range indicates robust bandgap stability, while the gradual frequency shift of band edges is attributed to temperature-induced changes in elastic constants and density of the constituent materials. Fig 11c highlights this effect through broadened transmission features, where the widening and displacement of resonance peaks formed at normalized frequency of 2.6 and 3.6, reflect the thermal modulation of acoustic impedance contrast. Collectively, these results demonstrate that Fibonacci quasi-periodicity provides enhanced tunability and thermal robustness of PnBG, which is advantageous for temperature-sensitive acoustic filtering and sensing applications.

The temperature-dependent transmission behavior of a quasiperiodic PnC composed of tungsten and polycrystalline silicon arranged according to the third-order Thue–Morse sequence is presented in Fig 12. In Fig 12a, the 3D transmission spectrum reveals well-defined stopbands and passbands whose spectral positions evolve with normalized frequency and temperature. The deterministic aperiodicity of the Thue–Morse ordering enhances multiple scattering and phase interference, leading to the formation of complex bandgap structures. Fig 12b highlights the emergence of narrow resonance peaks within the central region of the bandgap over specific temperature ranges. These resonances originate from thermally induced modifications in elastic constants and density, which enable localized acoustic modes to couple

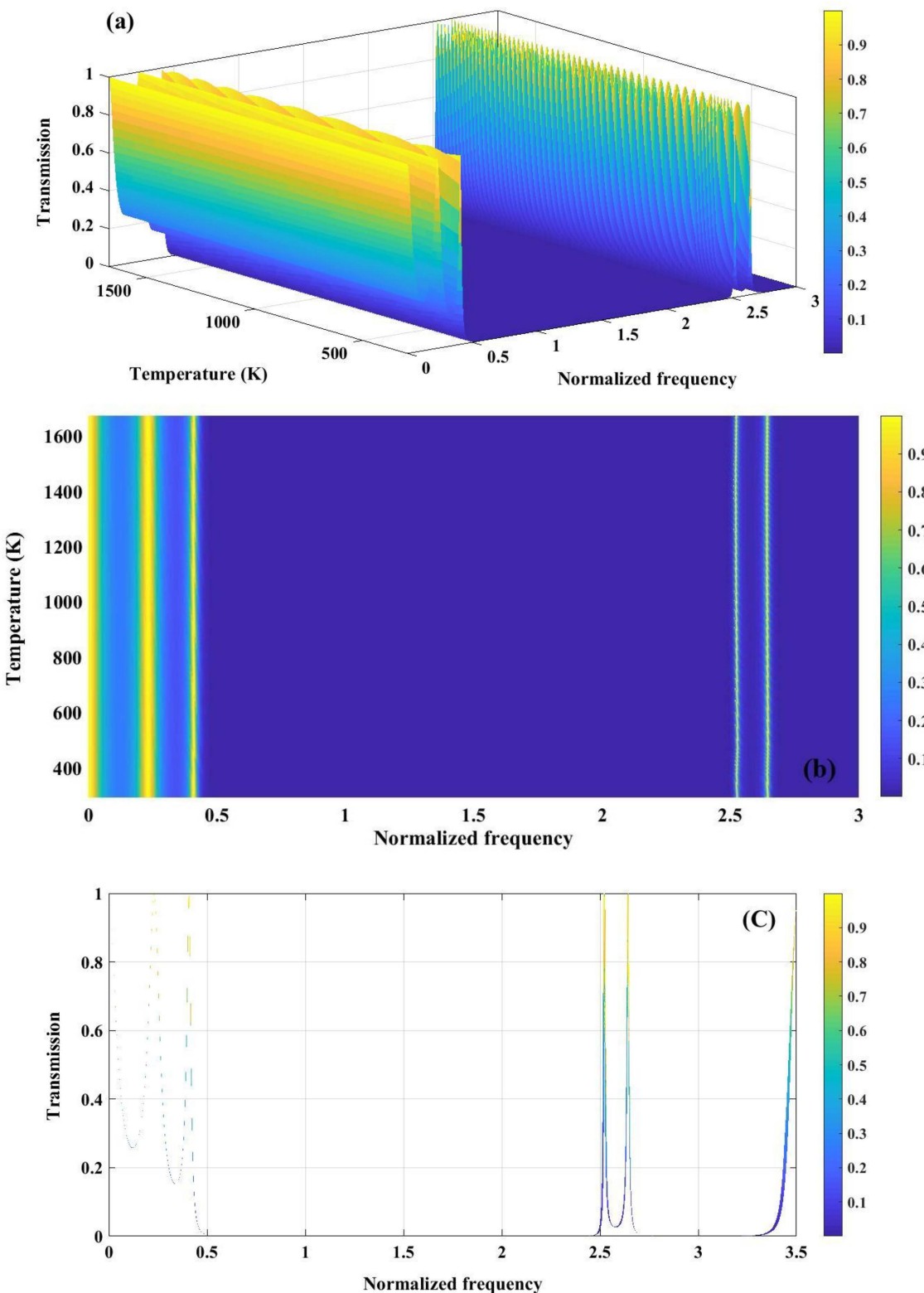

**Fig 10. (a) Analysis of transmission spectrum of a perfect PnC (Tungsten/ Polycrystalline silicon)[3] as a function of normalized frequency and temperature (b) variation in the band gaps in response to changes in temperature or normalized frequency (c) transmission spectrum versus normalized frequency.**

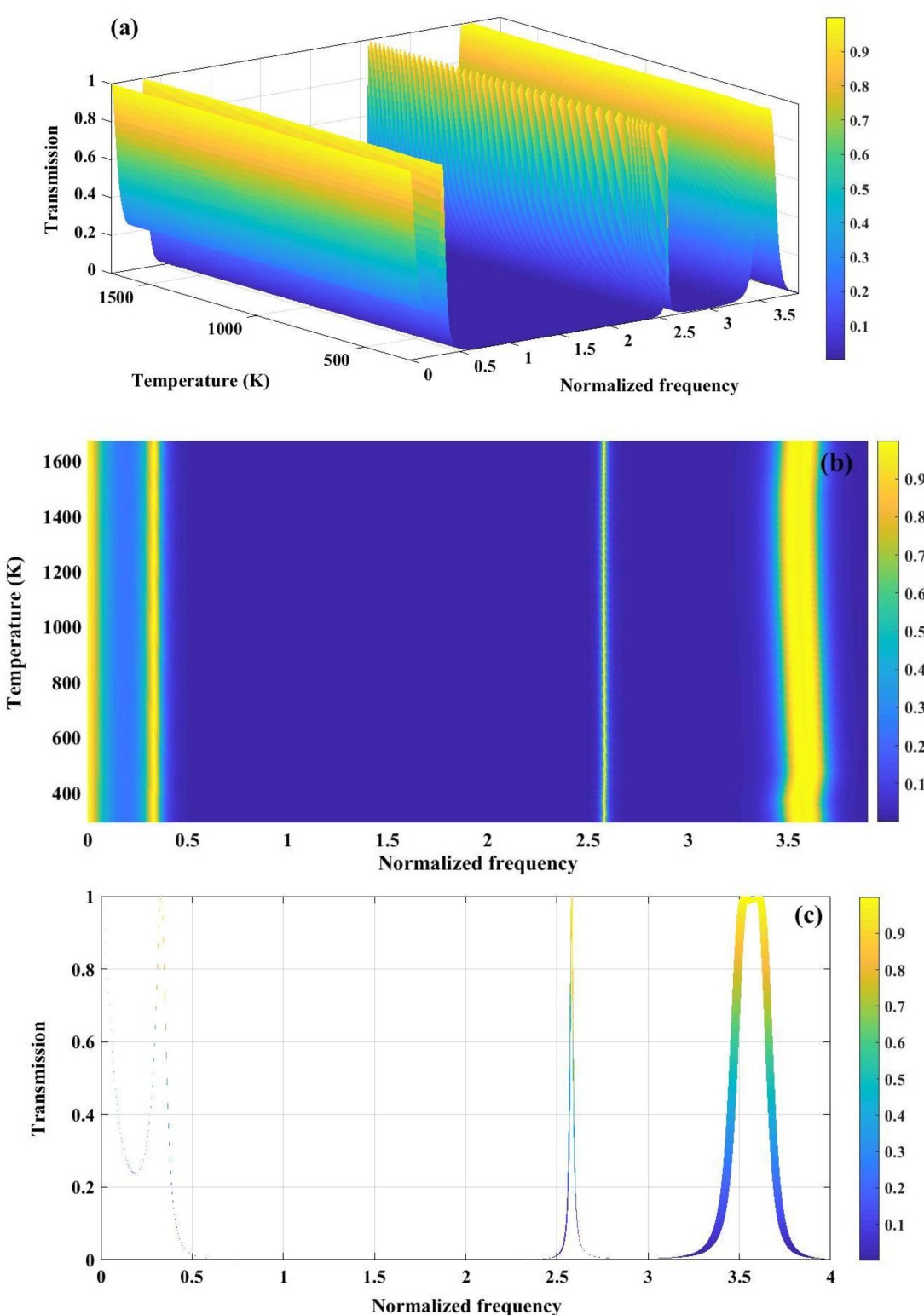

**Fig 11. (a) Transmission spectrum of a phononic crystal under three sequence of Fibonacci series configuration (Tungsten/ Polycrystalline silicon/ Tungsten).** (b) 2D colormap plot showing variation of temperature with respect to varying temperature (c) The broad line represented in the transmission graph reflects the changes in band gap width because of temperature fluctuations.

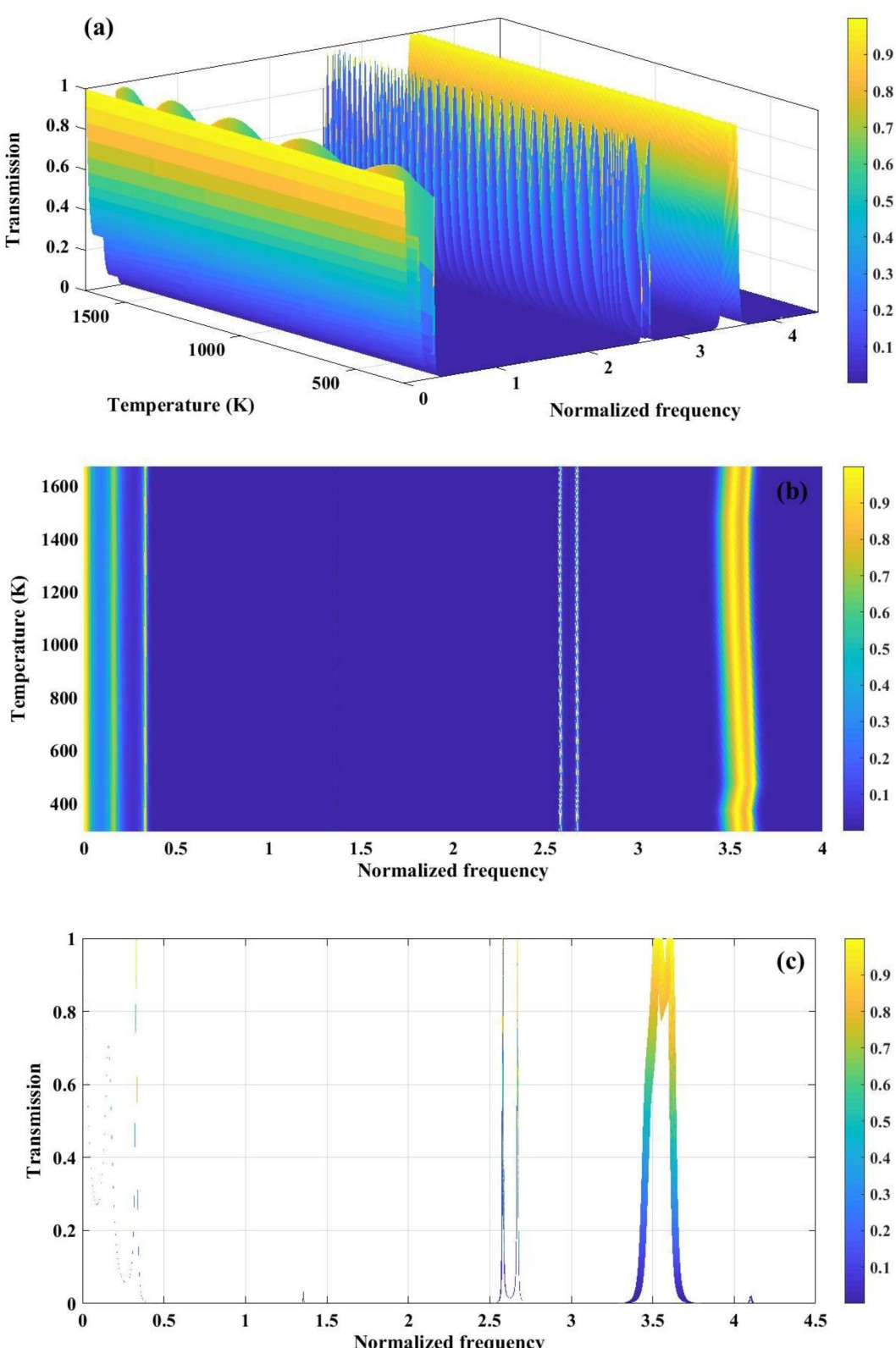

**Fig 12.** (a) the transmission spectrum of a phononic crystal, consisting of tungsten and polycrystalline silicon arranged based on the three sequence of the Thue-Morse series, is demonstrated, (b) the graph depicts the temperature ranges that induce the generation of resonance

**peaks within the central region of the band gap and (c) The wide line depicted in the transmission graph illustrates the variations in band gap width caused by changes in temperature.**

within the otherwise forbidden frequency region. Fig 12c further illustrates this effect through the broadening of transmission features, indicating temperature-driven variations in bandgap width. The observed widening and frequency shift of the band edges confirm the strong sensitivity of the Thue–Morse quasiperiodic structure to thermal modulation. Both the Fibonacci-structured configuration and the Thue-Morse arrangement show a band gap width of roughly 0.5–3.5 normalized frequency. Furthermore, resonance peaks are noticeable in the central area of the band gaps across the entire proposed temperature spectrum of 293–1673 K, thereby reinforcing the parallels between the two structures being analyzed. Largely, the figure demonstrates that Thue–Morse ordering provides enhanced control over temperature-tunable bandgaps and localized resonant modes, which is highly beneficial for adaptive phononic sensing applications.

The data presented in Fig 8 clearly illustrates that the transmission spectrum of a PnC designed with three sequences of double-periodic arrangements at a specific temperature, can yield satisfactory bandgaps. These bandgaps serve as a crucial basis for the application of this structure in the fields of detection and sensing. It is important to note that achieving a wide bandgap across the entire range of considered temperatures poses significant challenges. As depicted in Fig 13a, resonance peaks are observed to be localized at various temperatures within the middle of the bandgaps. Consequently, concentrating on a particular temperature and utilizing this structure as a dedicated temperature sensor can result in enhanced performance. Fig 13b and c offer a clear illustration that supports the established information. It demonstrates that, in addition to the temperature range responsible for irregularities in the transmission spectrum observed in earlier structures, localized resonance peaks emerge at both lower and higher temperature extremes in this instance. The final sequence examined in this study is the Cantor sequence. According to the definition of the sequence generation function, the structure is constructed with a polycrystalline silicon layer in the center that possesses a thickness nine times greater than that of a standard polycrystalline silicon cell.

The expected temperature fluctuations lead to unique conditions concerning band gaps and resonance peaks within the structure derived from the Cantor set, similar to previously analyzed structures. As depicted in Fig 14a, the transmission spectrum, plotted as a function of temperature across various normalized frequencies, highlights multiple band gaps within the commonly observed normalized frequency ranges identified in earlier studies. The substantial number of unit cells involved in the design of the structure results in pronounced band gaps characterized by sharp walls and distinct resonance peaks. The presence of resonance peaks within each band gap creates opportunities for utilizing the structure as a detector within a limited frequency range. Similar to double periodic structures, a specialized temperature sensor can enhance performance and broaden the potential applications of the engineered crystal in sensor technology and detection systems. In Fig 14b, the contour map reveals distinct temperature ranges where sharp resonance peaks emerge within the central region of the PnBG. These resonances originate from thermally induced modifications in elastic moduli and density, which enable localized acoustic modes supported by the fractal Cantor geometry to couple into the forbidden frequency region. The hierarchical, self-similar structure enhances multi-scale Bragg scattering, leading to strong wave localization. Fig 14c presents the transmission spectrum as a function of normalized frequency, clearly demonstrating the presence of multiple fragmented bandgaps. The appearance of several narrow transmission peaks interspersed between stopbands reflects the fractal nature of the Cantor sequence, which introduces band splitting and scale-dependent interference effects. Together, these results confirm that Cantor quasiperiodicity enables multi-bandgap formation and temperature-tunable resonant behavior, offering enhanced flexibility for acoustic sensing, and wave control applications.

## 6. Sensing performance analysis of the structures

A wide bandgap formation in the proposed quasi-periodic PnC structures reflect their capability and potential significance in sensing applications. Sensitivity serves as one of the tools that can aid in comparing different PnC structures to identify

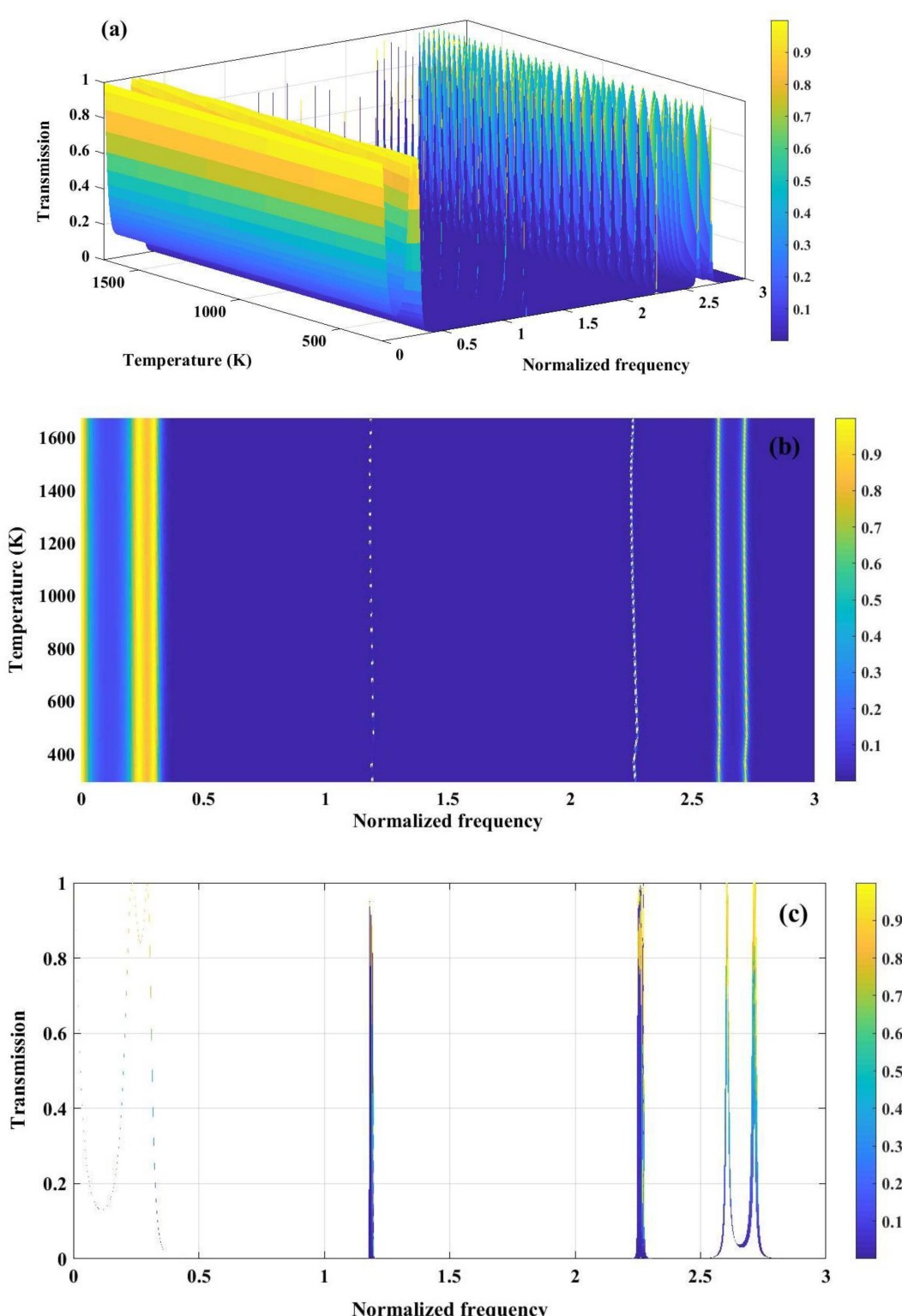

**Fig 13. (a) The transmission spectrum of a phononic crystal, composed of tungsten and polycrystalline silicon, is presented under optimal conditions and organized according to a three-sequence double-periodic series, (b) the graph illustrates the temperature intervals that trigger the formation of resonance peaks in the central area of the band gap and (c) the graph illustrates the transmission spectrum in relation to frequency ranges.** The broad line shown in the transmission graph represents the fluctuations in band gap width resulting from temperature variations.

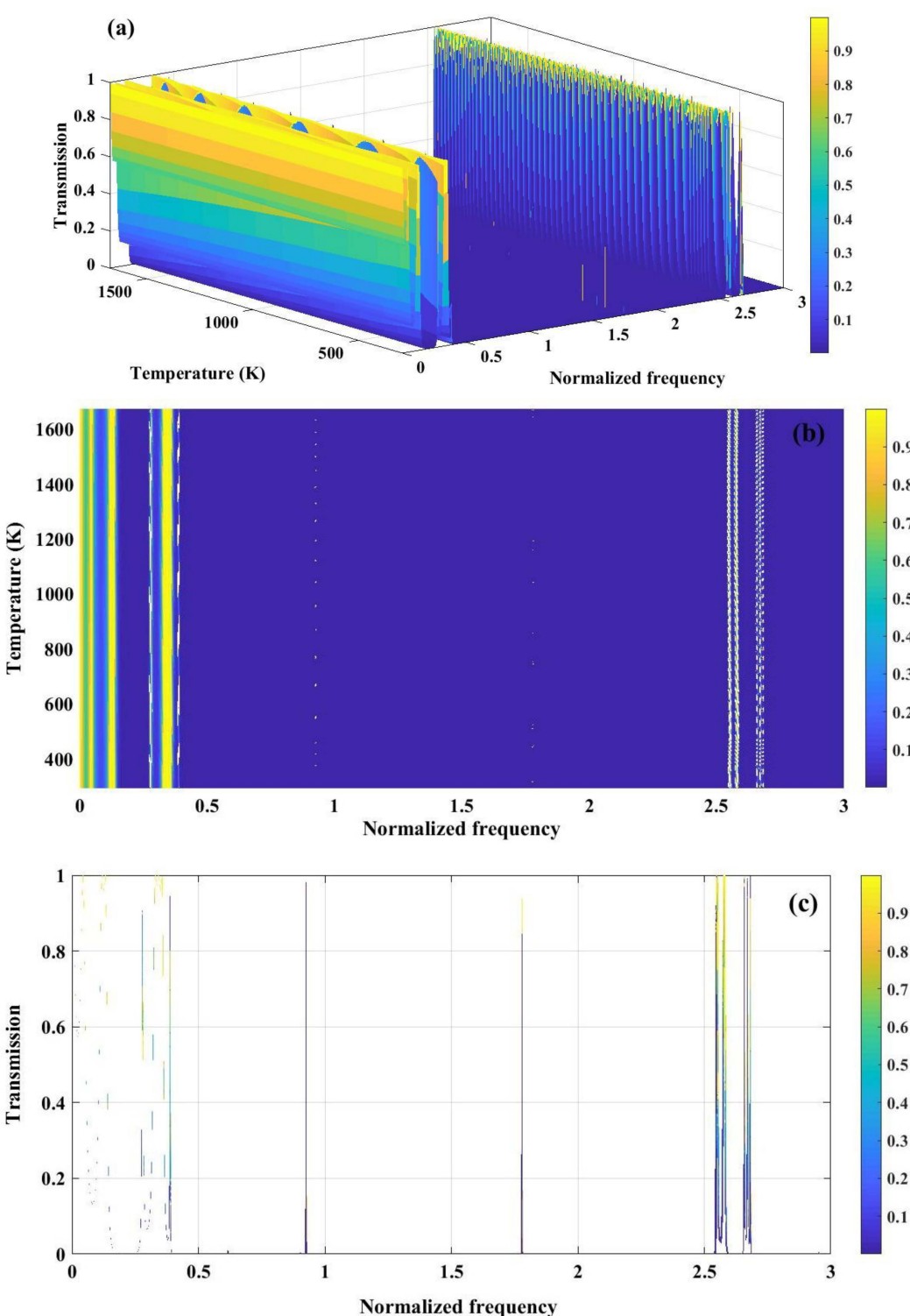

**Fig 14.** **(a) the three-dimensional perspective of the transmission spectrum for a phononic crystal made from tungsten and polycrystalline silicon, arranged according to the three sequences of the Cantor series, (b) the graph depicts the temperature ranges that induce the emergence of resonance peaks within the central region of the band gap and (c) the transmission spectrum in relation to normalized frequency reveals multiple bandgaps, which are observable in a two-dimensional representation of transmission as a function of normalized frequency.**

the most suitable configuration. Sensitivity is defined as the ratio of change in the band gap width to the temperature variations, which is expressed as,

$$S = \frac{W_f - W_i}{T_f - T_i}$$

(19)

Sensitivity is computed by considering bandgap width at 293K as a baseline reference. Based on the color maps provided in the preceding section, the temperature range deemed suitable for sensitivity calculations across all configurations spans from 293 K to 1673 K. The focus of sensitivity calculations is primarily on the width of the band gap. Fig 15 illustrates the sensitivity of different configurations in response to temperature fluctuations. The provided graphs depict the sensitivity characteristics of one-dimensional PnC across different structural configurations, encompassing both perfect and quasiperiodic arrangements. The sensitivity graph associated with the perfect periodic structure exhibits a uniform and stable sensitivity response throughout the examined temperature range. This consistent sensitivity progression positions the ideal structure as a preferred option for applications that necessitate stable and predictable sensitivity behavior. The sensitivity inherent in quasiperiodic configurations, including Fibonacci, Thue-Morse, double periodic, and Cantor arrangements, manifests in diverse and significant fluctuations. For example, the Fibonacci structure demonstrates comparatively smoother trends when compared with other quasiperiodic configurations, however, its sensitivity oscillations remain distinct from the stability observed in a perfect structure. The sensitivity graphs related to Cantor and Thue-Morse structures reveal notable irregularities in their sensitivity profiles, which are marked by distinct peaks and troughs at certain temperature ranges. These patterns highlight the inherent unpredictability of quasiperiodic structures in various applications. The double-periodic configuration represents an intermediary phase between periodic and quasiperiodic structures, displaying periodic traits that offer an enhanced stability equilibrium compared to the Cantor and Thue-Morse graphs. However, it falls short of attaining the uniform behavior typical of the perfect periodic graph. In a comparative analysis of the sensitivity graphs, the perfect periodic configuration is characterized by its predictability and stability in sensitivity, which are crucial

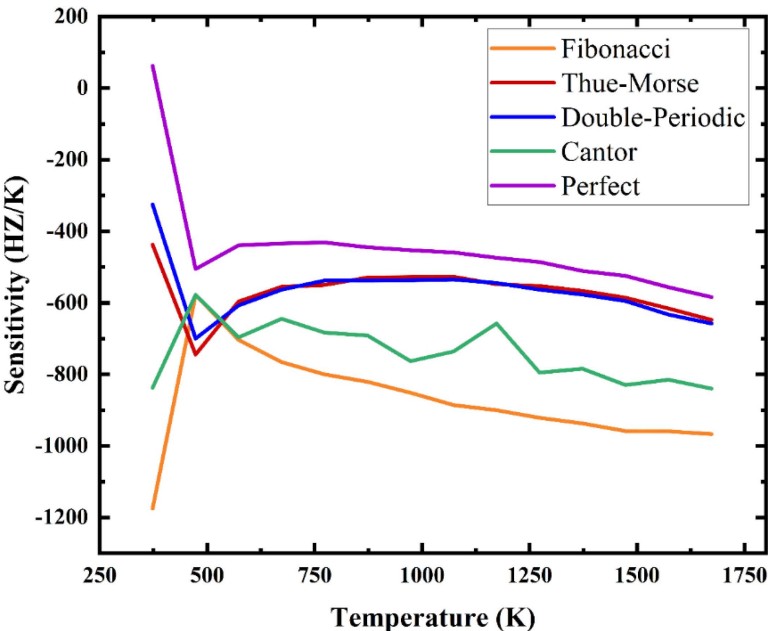

**Fig 15. The sensitivity of the designed structures versus temperature.**

for ensuring consistent sensitivity performance. In contrast, the quasiperiodic configurations offer distinct advantages, particularly the heightened sensitivity peaks observed in Cantor and Fibonacci structures. These features appear to be beneficial in specific scenarios that require enhanced sensitivity responses within designated temperature ranges. Fig 16 presents a range of perspectives and insights regarding the sensitivity of different configurations, their potentialities, and possible applications.

In Fig 16, the x-axis illustrates the sensitivity magnitudes, ranging from −1200 Hz/K to 200 Hz/K, while the y-axis displays the different temperature data points employed in the sensitivity assessments. This graph delineates the distribution of sensitivity magnitudes, which are derived from the temperature data points across five distinct configurations. As depicted in Fig 16 and S1 Table in the supporting information, the perfect configuration demonstrates a remarkable degree of uniformity, with sensitivity values primarily clustered around −500 Hz/K. This concentration of sensitivity around a particular value highlights its stable and predictable reaction to thermal variations. Such dependability renders it an optimal choice for applications that necessitate consistent performance in the face of temperature variations. Conversely, the quasiperiodic configurations exhibit distinct sensitivity distributions, emphasizing their unique thermal response characteristics. Of particular note, the Fibonacci configuration features a pronounced peak near −1000 Hz/K, indicating a significant concentration of sensitivity values within this specific range. The Thue-Morse configuration demonstrates a peak sensitivity of approximately −600 Hz/K, indicating a moderate degree of variability. In contrast, the Double-Periodic configuration reveals sensitivity values predominantly clustered around −500 Hz/K, which partially coincide with those of the Perfect configuration, suggesting a degree of commonality despite its quasiperiodic characteristics. Finally, the Cantor configuration shows a peak sensitivity close to −800 Hz/K, highlighting its unique response to temperature changes in comparison to other quasiperiodic structures. As illustrated in Fig 16, several fitted curves are superimposed on the graph. These curves offer significant statistical insights into the trends observed within the sensitivity distributions, facilitating a clearer and more straightforward comparison between periodic and quasiperiodic structures. In summary, this graph plays a crucial role in enhancing the understanding of how periodicity and quasi-periodicity affect the thermal behavior of the

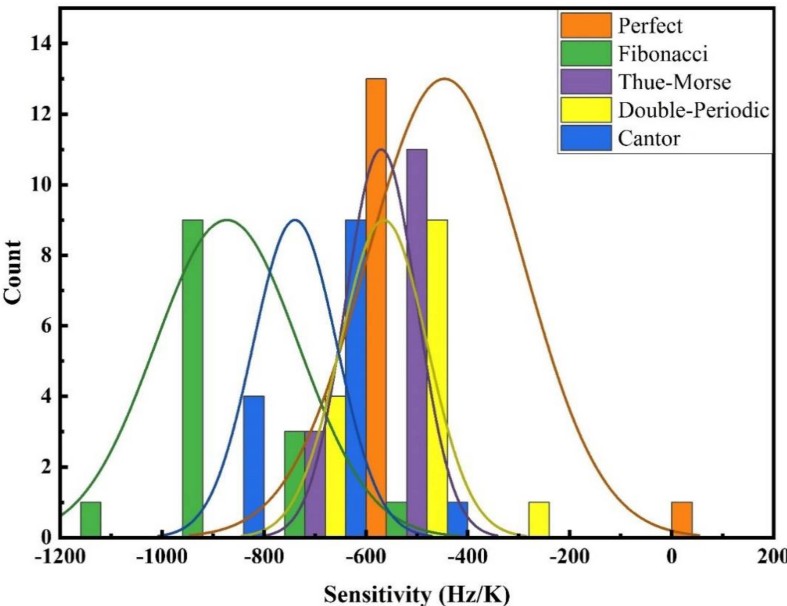

**Fig 16. The statistical distribution of sensitivity magnitudes derived from 14 temperature data points, with sensitivity calculated for each configuration.**

1D PnC structure. The pronounced differences in the behavior of these configurations underscore the distinct and specific characteristics of periodic and quasiperiodic PnCs. From both visual and quantitative perspectives, the differentiation of the sensitivity traits of these configurations highlights the respective advantages and limitations inherent to each type. Such insights are essential for optimizing the design and applications of PnC across various domains, including wave control, thermal sensing, and advanced materials science. The provided graphs and the information discussed above create a compelling representation of the performance of the designed structures, highlighting their capabilities and constraints. Based on the color maps and the calculated sensitivity, the comparison among the examined structures simplifies the process of selecting the optimal configuration. The following Table 1 presents concise and significant information regarding the various configurations.

It's worth noting that the negative sensitivity values indicate that the bandgap width decreases as temperature increases. Physically, this is attributed to thermal softening, where increasing temperature reduces the elastic modulus and slightly alters density, leading to a decrease in acoustic wave velocity and impedance contrast. Consequently, the band edges shift toward lower frequencies, resulting in a reduced bandgap width. Thus, negative sensitivity reflects an inverse temperature-response relationship, not a degradation in performance.

Lastly, we introduced a normalized performance metric (figure of merit, FOM) that combines both the bandgap width and the stability of sensitivity. The bandgap width was normalized with respect to the maximum observed value, while stability was quantified using the inverse (or normalized form) of the standard deviation of sensitivity over the considered temperature range. Specifically, the combined metric is defined as:

$$FOM_i = \alpha \frac{B_i}{B_{max}} + (1 - \alpha)(1 - \frac{\sigma_i}{\sigma_{max}})$$

(20)

where $B_i$ is the bandgap width, $\sigma_i$ is the standard deviation of sensitivity, $B_i$ is the bandgap width of structure $I$, $B_{max}$ is the maximum bandgap among all structures, and $\alpha$ is a weighting factor (taken as 0.5 for equal importance). Also, the normalized bandgap width $B^*$ is given as; $B^* = \frac{B_i}{B_{max}}$. In addition, the stability of sensitivity is given as $S^* = 1 - \frac{\sigma_i}{\sigma_{max}}$.

This formulation enables an objective comparison between structures by simultaneously accounting for performance (bandgap width) and robustness (stability). Based on this metric, while the Fibonacci structure exhibits the largest bandgap, its relatively higher sensitivity fluctuations reduce its overall score. In contrast, the periodic structure shows excellent stability but a smaller bandgap.

Interestingly, the double-periodic structure emerges as an optimal compromise, achieving a balanced combination of reasonably wide bandgaps and improved stability compared to other quasiperiodic configurations. All normalized metrics of the designs are computed based on Equation (20) and all inserted in Table 2.

## 7. Manufacturing inaccuracies

Theoretical studies play a crucial role in designing optimal and practical PnC devices, as simulations often show strong agreement with experimental results. However, fabrication-induced imperfections such as layer thickness deviations,

**Table 1. Performance outputs of the different PnC structures.**

| Configuration | Maximum sensitivity (Hz/K) | Maximum Bandgap width (Hz) | Special feature |
|---|---|---|---|
| Periodic | −584.05 | $11.889 \times 10^6$ | Broad stable band gaps |
| Fibonacci | −1175 | $18.094 \times 10^6$ | Localized sharp resonance peaks |
| Thue-Morse | −744.45 | $12.684 \times 10^6$ | Two coupled resonance peaks |
| Double-Periodic | −700 | $13.021 \times 10^6$ | Applicable for dedicated temperature sensor |
| Cantor | −577.77 | $15.342 \times 10^6$ | sharp walls and distinct resonance peaks. |

**Table 2. Normalized Metrics of the performance of all suggested designs.**

| Structure | Bandgap ($B^*$) | Stability ($S^*$) | $FOM_i$ |
|---|---|---|---|
| Periodic | 0.657 | 0.503 | 0.580 |
| Fibonacci | 1.000 | 0 | 0.500 |
| Thue-Morse | 0.701 | 0.366 | 0.534 |
| Double-period | 0.720 | 0404 | 0.562 |
| Cantor | 0.848 | 0.508 | 0.678 (BEST) |

surface roughness from machining, and material voids or impurities can alter Young's modulus and density, leading to deviations from ideal theoretical predictions. Ignoring these effects often result in inaccurate conclusions. Monte Carlo (MC) simulations provide a robust statistical framework to evaluate such uncertainties by introducing controlled random variations in key structural parameters. This numerical approach realistically models fabrication tolerances and assesses their impact on PnBG characteristics across real-world engineering applications.

In this section, a Monte Carlo simulation is conducted using randomly sampled values for the thicknesses of the layers, Young's modulus, and material densities, incorporating a 5% error margin for the perfect periodic structure. The general Monte Carlo framework can be expressed as,

$$X_i = X_0 + \sigma_X.\xi_i \qquad \xi_i \sim N(0, 1) \tag{21}$$

Where, $X_i$ represents the varied parameter in the ith realization, $X_0$ is the nominal value, $\sigma_X$ denotes to the standard deviation of the variation, which is 5% here, and $\xi_i$ is the Gaussian-distributed noise.

$$\mu\Delta f = \frac{1}{N}\sum_{i=1}^{N}\Delta f_i \tag{22}$$

$$\sigma\Delta f = \sqrt{\frac{1}{N}\sum_{i=1}^{N}(\Delta f_i - \mu\Delta f)^2} \tag{23}$$

Upon examining Fig 17a, it becomes evident that the transmission spectrum derived from multiple Monte Carlo simulations, accompanied by a shaded area around the mean curve, offers valuable insights into the manufacturability of the proposed structure and suggests potential design adjustments. This shaded area illustrates the standard deviation (σ) of transmission at each frequency, shedding light on the sensitivity of the PnC's bandgap structure to fabrication errors. In essence, the standard deviation measures the extent to which the transmission response varies due to random fluctuations in material properties and geometric parameters. In the depicted figure, certain frequency ranges exhibit broad shaded areas, which signify a heightened sensitivity to fabrication inaccuracies. This indicates that even minor variations in thickness or material characteristics can lead to significant changes in the transmission properties. This can lead to a fluctuation or shifting of bandgaps and making the structure less predictable and reliable in practical applications. In contrast, narrow shaded regions highlighting a robust bandgap structure that is more tolerant to manufacturing variations. In other word, the supposed designed structure response remains relatively stable despite fabrication imperfections. Overall, the findings from the Monte Carlo analysis offer critical insights into the manufacturability of the proposed structure and suggest possible design adjustments to improve its durability. In Fig 17b, a histogram analysis is presented, derived from the transmission spectra obtained through various Monte Carlo simulations. The characteristics of the histogram's distribution, including its

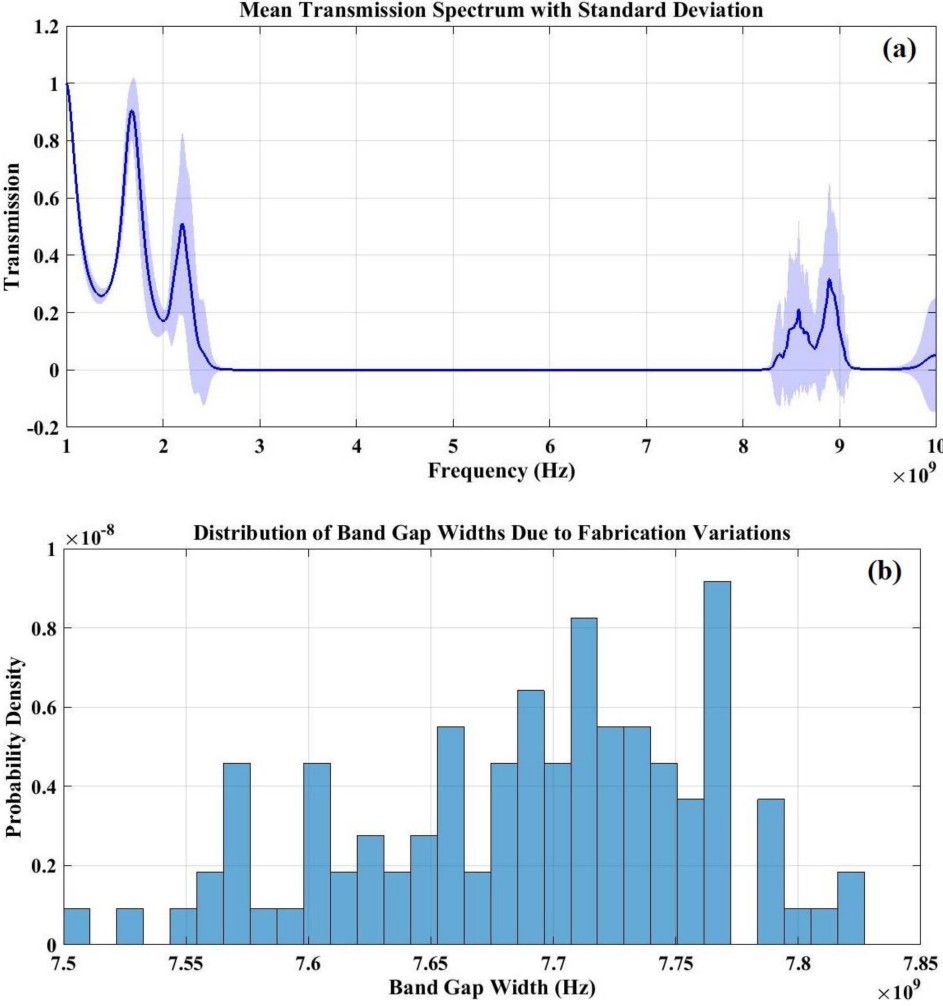

**Fig 17. (a) Monte Carlo Simulation depicting manufacturing inaccuracies for an ideal periodic structure configured at a temperature of 293 K.** This simulation incorporates a 5% random variation in the thickness of layers, densities, and Young's modulus and (b) histogram graphs from the Monte Carlo simulation, which illustrate the statistical distribution of transmission values at a specified frequency.

shape and spread, serve as critical indicators of the sensitivity to fabrication processes. In contrast to the mean transmission spectrum, which reflects an averaged response, the histogram emphasizes the frequency of different transmission values, enabling an evaluation of the statistical variation in transmission for a specific PnC configuration. A histogram that displays a narrow and well-defined peak suggests that, despite random fluctuations in the intrinsic properties of materials and geometric parameters, the transmission response remains stable across different iterations. This behavior signifies a low sensitivity to fabrication inaccuracies, indicating that the designed PnC structure is both robust and dependable for practical applications. On the other hand, a wider histogram distribution indicates considerable variability in transmission, suggesting a heightened sensitivity to changes in layer thickness, density, and elastic properties.

## 8. Fabrication feasibility

The current design with millimeter-scale layers is intended for proof-of-concept and macroscale validation, where fabrication can be achieved using conventional techniques such as precision machining, layer stacking, diffusion bonding, or

sputter deposition followed by wafer bonding [57]. For practical sensing applications, especially MEMS integration, the structure can be scaled down proportionally since PnBGs depend on the ratio of wavelength to lattice constant. Numerous thin-film deposition like chemical vapor deposition (CVD) and physical vapor deposition (PVD), lithography, and etching can be employed to realize microscale multilayers [58,59]. So, with these advanced methodologies, it is feasible to fabricate the proposed design towards MEMS-compatible platforms.

However, we acknowledge that practical implementation may be constrained by thermo-mechanical stresses, interfacial diffusion, and oxidation, particularly for polycrystalline silicon and tungsten multilayers at elevated temperatures. In realistic applications, the operational range would be lower unless protective coatings of oxide or nitride layers, vacuum environments, and advanced high-temperature fabrication techniques are employed.

## 9. Conclusion

This work systematically compares periodic and quasiperiodic PnC architectures composed of tungsten and polycrystalline silicon to evaluate their sensing performance under varying temperature conditions. Transfer matrix technique is employed to study the transmittance spectrum of the designed PnC structures. Simulation upshots revealed that the ideal three-unit-cell periodic structure serves as a robust benchmark, exhibiting uniform bandgap characteristics and a stable, linear sensitivity response over the investigated temperature range, with a maximum sensitivity of 62.5 Hz/K at 373 K. In contrast, quasiperiodic configurations, particularly the Fibonacci sequence, enable enhanced wave localization and achieve a widened bandgap of $18 \times 10^6$ Hz at 373K, along with pronounced resonance-induced sensitivity enhancement. However, the aperiodic layer arrangement introduces spectral fluctuations and reduced stability, which may limit their suitability for applications requiring repeatable performance. Monte Carlo simulations further demonstrate the influence of fabrication tolerances on transmission characteristics, emphasizing the necessity of statistical robustness analysis. Overall, structure selection should be guided by the required trade-off between sensitivity enhancement and operational stability.

## Supporting information

**S1 Table. The statistical distribution of sensitivity magnitudes derived from 14 temperature data points, with sensitivity calculated for each configuration.**
(DOCX)

## Author contributions

**Conceptualization:** Farhad Javanpour Heravi.

**Data curation:** Farhad Javanpour Heravi, Hussein A. Elsayed.

**Formal analysis:** Farhad Javanpour Heravi.

**Funding acquisition:** May Bin-Jumah, Haifa A Alqhtani, Mostafa R. Abukhadra.

**Investigation:** Ahmed Mehaney.

**Methodology:** Ahmed Mehaney.

**Project administration:** May Bin-Jumah, Haifa A Alqhtani.

**Resources:** Ahmed Mehaney.

**Software:** Farhad Javanpour Heravi, Ahmed Mehaney.

**Supervision:** Hussein A. Elsayed, Ali Hajjiah, May Bin-Jumah, Haifa A Alqhtani, Mostafa R. Abukhadra, Emad Solouma, Suryakanta Nayak, Wail Al Zoubi, Ahmed Mehaney.

**Validation:** Hussein A. Elsayed, Ahmed Mehaney.

**Visualization:** Ahmed Mehaney.

**Writing – original draft:** Farhad Javanpour Heravi, Ahmed Mehaney.

**Writing – review & editing:** Hussein A. Elsayed, Ali Hajjiah.

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
