## [Decision Letter · Decision Letter 0]

8 Mar 2026

PONE-D-26-00953Comparative Analysis of Temperature Effect on Bandgap Characteristics in 1D Phononic Crystals: Periodic Versus Quasiperiodic structuresPLOS One

Dear Dr. A. Elsayed,

Thank you for submitting your manuscript to PLOS ONE. After careful consideration, we feel that it has merit but does not fully meet PLOS ONE’s publication criteria as it currently stands. Therefore, we invite you to submit a revised version of the manuscript that addresses the points raised during the review process.

We look forward to receiving your revised manuscript.

Kind regards,

Talaat Abdel Hamid, Ph.D

Academic Editor

PLOS One

Journal Requirements:

[The authors acknowledge Princess Nourah bint Abdulrahman University Researchers Supporting Project number (PNURSP2025R737), Princess Nourah bint Abdulrahman University, Riyadh, Saudi Arabia.].

[The authors acknowledge Princess Nourah bint Abdulrahman University Researchers Supporting Project number (PNURSP2025R737), Princess Nourah bint Abdulrahman University, Riyadh, Saudi Arabia.]

[The authors acknowledge Princess Nourah bint Abdulrahman University Researchers Supporting Project number (PNURSP2025R737), Princess Nourah bint Abdulrahman University, Riyadh, Saudi Arabia.]

6. We note that the captions for your main Figures and Tables are similar on what is indicated in the Supporting Information file.

Please confirm the intended placement for these captions. If these are indeed part of the main manuscript, kindly remove the file with “Supporting Information” captions to prevent redundancy.

Reviewers' comments:

Reviewer's Responses to Questions

**Comments to the Author**

1. Is the manuscript technically sound, and do the data support the conclusions?

Reviewer #1: No

Reviewer #2: Yes

Reviewer #3: Yes

2. Has the statistical analysis been performed appropriately and rigorously? 

Reviewer #1: N/A

Reviewer #2: I Don't Know

Reviewer #3: Yes

3. Have the authors made all data underlying the findings in their manuscript fully available?

Reviewer #1: Yes

Reviewer #2: Yes

Reviewer #3: No

4. Is the manuscript presented in an intelligible fashion and written in standard English?

Reviewer #1: Yes

Reviewer #2: Yes

Reviewer #3: Yes

5. Review Comments to the Author

Reviewer #1:

1. Please review the text to ensure that acronyms are not defined twice (e.g., PnC). Also, once an acronym is defined, it should be used (e.g., PnBG).

2. There is a vast literature on quasi-periodic phononic crystals. The presented literature review should position this work relative to the existing ones.

3. The historical background in the Introduction does not seem necessary for the context of the presented research.

4. The authors present a realization with “three unit cells”. However, when discussing band diagrams, one refers to periodic structures. Why presenting a finite structure at this point?

5. “In the analysis of the transmission spectrum, Young’s modulus, density, and shear stress are critical

parameters, as illustrated in figure 1”. Figure 1 does not deal with material parameters.

6. The authors explicitly mention the Poisson’s ratios for the two considered materials, but other material properties are not textually described.

7. The authors should include references for the interested reader concerning basic phenomena involving band gap formation (e.g., Bragg scattering) and the formulation of the mathematical model for obtaining transmission spectra.

8. Eq. (1) should be a part of a wider system of equations (including other directions). It is not clear what assumptions concerning the types of propagating waves and boundary conditions the authors have assumed.

9. Both lambda and mu refer to Lame parameters. However, mu is also recognized as the shear modulus (not shearing). Please verify along the manuscript if there are other typos.

10. After Eq. (9), the authors mention a component of shear stress. But so far, the presented equations referred to the propagation of longitudinal waves. Please clarify.

11. The authors mention “angle of incidence”, but the presented structure is a waveguide. Therefore, what does this angle refer to? If this involves incident acoustic waves, this is not clear from the discussion. In this case, also, the numerical methods utilized in the work should be discussed.

12. Where was the data from Fig. 2 obtained from? Did the authors use a reference or measure it?

13. Figs. 3 and 4 seem to represent a parametric sweep (not an optimization, which involves setting an objective function, constraints, etc.).

14. Figures 6-9 are only qualitatively described, but this should also be done quantitatively.

Reviewer #2: My Comments are given below:

1.The abstract initially refers to “magnetic field sensing,” whereas the study focuses on temperature sensing. Can the authors clarify this inconsistency and clearly define the primary sensing mechanism investigated?

2.Several studies have already explored periodic and quasiperiodic 1D phononic crystals for sensing applications. What is the clear quantitative advancement of this work beyond existing literature, particularly regarding bandgap enhancement and sensitivity improvement?

3.Why were tungsten and polycrystalline silicon specifically selected? Were alternative material combinations evaluated for higher acoustic impedance contrast or improved thermal stability?

4.The temperature range extends from 293 K to 1673 K. Considering material limitations (e.g., oxidation, phase stability), how realistic is this range for practical implementation of the proposed sensor?

5.The model assumes ideal interfaces and longitudinal wave propagation. How would interface roughness, mode conversion (shear waves), or thermoelastic damping affect the reported bandgap and sensitivity results?

6.Sensitivity is defined only based on bandgap width variation. Why was resonance frequency shift not considered, as it is commonly more precise for sensing applications?

7.The manuscript mentions 5% fabrication tolerances analyzed via Monte Carlo simulations.

How many iterations were performed?

Were statistical confidence intervals calculated?

How do these uncertainties quantitatively impact sensitivity?

8.Although Fibonacci shows the widest bandgap, periodic structures show more stable sensitivity. Could the authors provide a normalized performance metric combining both bandwidth and stability to identify the optimal design objectively?

9.Figure 16 shows negative sensitivity values (e.g., −500 Hz/K to −1000 Hz/K). What is the physical interpretation of negative sensitivity in this context?

10.The proposed structure involves millimeter-scale layers (e.g., up to 9 mm thickness).

What fabrication technique is envisioned?

Has scalability toward MEMS-compatible dimensions been considered?

Reviewer #3: This manuscript presents a comparative numerical investigation of temperature-dependent bandgap behavior in 1D tungsten/polycrystalline silicon phononic crystals, examining periodic and several quasiperiodic configurations (Fibonacci, Thue–Morse, double-periodic, and Cantor). The topic is relevant to acoustic wave engineering and sensing applications, and the attempt to systematically compare multiple deterministic aperiodic sequences under identical material platforms is commendable. The inclusion of fabrication tolerances and Monte Carlo–based robustness analysis adds practical value.

However, the manuscript in its current form requires substantial clarification and strengthening before it can be considered for publication. First, the novelty claim is somewhat overstated, as comparative studies of quasiperiodic phononic crystals for bandgap engineering are well represented in the literature; the authors should more clearly delineate what fundamentally new physical insight is provided beyond incremental parametric comparison. Second, the physical model lacks sufficient detail regarding boundary conditions, convergence testing, normalization procedures, and validation against either analytical limits or prior experimental/theoretical benchmarks. The temperature range up to 1673 K appears extremely high for polycrystalline silicon and tungsten multilayers, yet material stability and thermo-mechanical constraints are not discussed. The definition of sensitivity based solely on bandgap width variation may oversimplify sensing performance; resonance frequency shift or Q-factor analysis would provide a more rigorous metric. Additionally, the manuscript contains significant language issues, repetition, and organizational redundancy that obscure the key findings. Figures are extensively described but not critically interpreted in terms of physical mechanisms. Overall, while the study has potential, major revisions are necessary to clarify methodology, strengthen physical justification, moderate claims of novelty, and improve scientific rigor and presentation quality.

6. PLOS authors have the option to publish the peer review history of their article (what does this mean?). If published, this will include your full peer review and any attached files.

Reviewer #1: No

Reviewer #2: No

Reviewer #3: **Yes:**Abrar Islam

---

## [Author Response · Author response to Decision Letter 1]

1 Apr 2026

Authors' Response to Reviewers’ comments

Manuscript Number: PONE-D-26-00953

Title: Comparative Analysis of Temperature Effect on Bandgap Characteristics in 1D Phononic Crystals: Periodic Versus Quasiperiodic structures

Journal: PLOS One

To

The Editor,

Respected Professor,

The authors would like to sincerely thank the reviewers for their meticulous reading of the manuscript and making valuable suggestions for enriching and improving the quality and importance of the manuscript for the scientific community. Authors do believe that the revisions made in the final version of the manuscript have a significant contribution from such sincere and constructive review of the reviewer. Therefore, in submission of gratitude and respect, authors acknowledge the anonymous reviewer. The comments and suggestions are highly appreciated and included in the modified version of the manuscript. All the changes made by the authors are highlighted in red color within the revised manuscript. Also, itemized responses to the specific questions/queries raised by the reviewers are presented through the uploaded file named "Response to Reviewers", for clarification.

With best regards:

Prof: Hussein A. Elsayed

The corresponding Author

---

## [Decision Letter · Decision Letter 1]

8 May 2026

Comparative Analysis of Temperature Effect on Bandgap Characteristics in 1D Phononic Crystals: Periodic Versus Quasiperiodic structures

PONE-D-26-00953R1

Dear Dr. A. Elsayed,

We’re pleased to inform you that your manuscript has been judged scientifically suitable for publication and will be formally accepted for publication once it meets all outstanding technical requirements.

Kind regards,

Talaat Abdel Hamid, Ph.D

Academic Editor

PLOS One

Additional Editor Comments (optional):

Reviewers' comments:

Reviewer's Responses to Questions

**Comments to the Author**

1. If the authors have adequately addressed your comments raised in a previous round of review and you feel that this manuscript is now acceptable for publication, you may indicate that here to bypass the “Comments to the Author” section, enter your conflict of interest statement in the “Confidential to Editor” section, and submit your "Accept" recommendation.

Reviewer #1: (No Response)

Reviewer #2: All comments have been addressed

Reviewer #3: All comments have been addressed

2. Is the manuscript technically sound, and do the data support the conclusions?

Reviewer #1: Partly

Reviewer #2: Yes

Reviewer #3: Yes

3. Has the statistical analysis been performed appropriately and rigorously? 

Reviewer #1: N/A

Reviewer #2: I Don't Know

Reviewer #3: Yes

4. Have the authors made all data underlying the findings in their manuscript fully available?

Reviewer #1: Yes

Reviewer #2: Yes

Reviewer #3: No

5. Is the manuscript presented in an intelligible fashion and written in standard English?

Reviewer #1: Yes

Reviewer #2: Yes

Reviewer #3: Yes

6. Review Comments to the Author

Reviewer #1: 1. The first comment concerns the definition of acronyms. The authors have replied to a distinct inquiry.

2. Nonetheless, an ideal structure does not consist of “three unit cells”, but rather, an infinite number.

3. Despite my previous comment, the authors still refer to the selection of “optimal” angles.

Reviewer #2: Author answered all the questions properly.The paper can be accepted in its present form for publication.

Reviewer #3: Authors have addressed the reviewers' comments well in the update version. The revised manuscript is acceptable as is.

7. PLOS authors have the option to publish the peer review history of their article (what does this mean?). If published, this will include your full peer review and any attached files.

Reviewer #1: No

Reviewer #2: No

Reviewer #3: **Yes:**Abrar Islam

---

## [Editor Report · Acceptance letter]

PONE-D-26-00953R1

PLOS One

Dear Dr. Elsayed,

I'm pleased to inform you that your manuscript has been deemed suitable for publication in PLOS One. Congratulations! Your manuscript is now being handed over to our production team.

Kind regards,

on behalf of

Dr. Talaat Abdel Hamid

Academic Editor

PLOS One